# The Effect of Meat and Bone Meal (MBM) on the Seed Yield and Quality of Winter Oilseed Rape

Aleksandra Załuszniewska * and Anna Nogalska 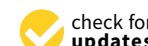

Department of Agricultural Chemistry and Environmental Protection, University of Warmia and Mazury in Olsztyn, 10-719 Olsztyn, Poland; anna.nogalska@uwm.edu.pl
* Correspondence: aleksandra.zaluszniewska@uwm.edu.pl; Tel.: +48-509408399

**Abstract:** The aim of a two-year field experiment conducted in north-eastern (NE) Poland was to evaluate the effect of meat and bone meal (MBM) applied without or with mineral nitrogen (N) on seed yield, thousand seed weight (TSW), protein yield, fat yield, fatty acid profile and glucosinolate (GLS) concentrations in winter oilseed rape. Five treatments were compared: MBM applied at 1.0, 1.5, 2.0 Mg ha$^{-1}$, inorganic NPK, and a zero-N check. The first two MBM plots and the NPK plots received supplemental inorganic N to provide a total of 158 kg N ha$^{-1}$. The yields of winter oilseed rape were highest in the treatment with mineral (NPK) fertilization. All plots receiving MBM yielded equally to each other but greater than the unfertilized check. Winter oilseed rape accumulated significantly more protein in seeds in the NPK treatment than in the 1.5 Mg ha$^{-1}$ MBM + 40 kg N ha$^{-1}$ treatment. The crude fat content of seeds was significantly higher in the 1.5 Mg MBM ha$^{-1}$ + 40 kg N ha$^{-1}$ treatment, compared with the NPK treatment and the 1.0 Mg MBM ha$^{-1}$ + 79 kg N ha$^{-1}$ treatment. Oleic, linoleic, and α-linolenic acids accounted for nearly 90% of total fatty acids in rapeseed oil, and the average ratio of linoleic acid to α-linolenic acid was 1.81:1. Fertilization had a minor influence on the proportions of fatty acids, which were considerably affected by adverse weather conditions.

**Keywords:** seed yield; protein; fatty acids; glucosinolates; fertilizer

## 1. Introduction

Oilseed crops of the genus Brassica are the third-leading source of vegetable oil in the world, after palm oil and soybean oil. Winter oilseed rape (*Brassica napus* L.) is the most important oilseed and protein crop in Europe, including Poland. Rapeseed accounts for around 80% and 95% of oilseed crops grown in Western Europe and Poland, respectively. Rapeseed has various industrial applications; it is grown for the production of edible vegetable oils and biofuels. The byproducts of rapeseed processing for oil production (cake and meal) are valuable high-protein components of animal concentrate feeds. Biological progress in oilseed rape breeding is an important consideration due to the growing market demand for rapeseed oil, observed particularly in the EU countries [1]. Modern breeding programs focus on improving the qualitative traits of rapeseed, including the concentrations and composition of fatty acids, protein content and fiber content, as well as stress tolerance. Breeding progress, in particular the development of hybrid varieties, has also contributed to a considerable increase in yields [2–6].

Winter oilseed rape has high nutrient requirements [7,8]. Fertilization is one of the key factors influencing the yield and quality of oilseed rape seeds [3]. In winter oilseed rape, seed quality is determined by the content of crude fat, total protein, antinutritional factors, and fatty acid profile [9]. Meat and bone meal (MBM) can be an alternative to organic and mineral (nitrogen—N and phosphorus—P) fertilizers. It is rich in N, P, Ca, micronutrients, and organic matter that can be recycled back to the soil. In MBM, N is present in the form of protein compounds, which is released into the soil through mineralization and becomes available to plants already in the first

year after application [10–12]. Nitrogen supplied by MBM can meet 80% of the N requirements of cereals [13] but the remaining 20% must be provided by mineral fertilizers. Phosphorus is present in MBM in organic form (meat fraction), which is available to plants, and in the form of hydroxyapatite (bone fraction), from which available P is released in an acidic environment [14,15]. Economically extractable phosphate rock reserves are expected to be depleted within the next few decades around the world because they constitute non-renewable natural resources that cannot be replaced. Therefore, the recovery and recycling of P from waste streams contribute to sustainable P use worldwide [16–18].

Previous field experiments [19–22] have demonstrated that only high doses of MBM supplied crops with N in adequate amounts, comparable with mineral fertilizers. However, the application of high doses of MBM leads to the accumulation of P in the soil; 2.5 Mg of MBM is equivalent to 110 kg P ha$^{-1}$, which considerably exceeds the P requirements of most crop plants [23,24]. In view of the above, the present study was conducted to test the hypothesis that lower doses of MBM supplemented with mineral N might raise the soil N:P ratio from 1.75 to values above 2.0, and exert a positive effect on the yields of seeds, protein and fat, fatty acid profile and the content of glucosinolates (GLS) in winter oilseed rape.

## 2. Materials and Methods

### 2.1. Experimental Site and Treatments

A field experiment with winter oilseed rape was conducted in two growing seasons of 2015/16 and 2016/17 at the Agricultural Experiment Station in Tomaszkowo located in north-eastern (NE) Poland, owned by the University of Warmia and Mazury in Olsztyn. The experimental treatments were as follows: (1) zero-fert—no fertilization; (2) inorganic NPK treatment (with mineral fertilization); (3) 1.0 Mg ha$^{-1}$ MBM + N (79 kg); (4) 1.5 Mg ha$^{-1}$ MBM + N (40 kg); (5) 2.0 Mg ha$^{-1}$ MBM (without mineral N) (Table 1).

**Table 1.** Rates of nitrogen (N), phosphorus (P) and potassium (K) applied with meat and bone meal (MBM) and mineral fertilizers (kg ha$^{-1}$) to winter oilseed rape in 2015–2017.

| Treatment | 2015/16 | | | 2016/17 | | |
|---|---|---|---|---|---|---|
| | N | P | K | N | P | K |
| 1. Zero-fert | 0 | 0 | 0 | 0 | 0 | 0 |
| 2. Inorganic NPK * | 158 | 45 | 145 | 158 | 45 | 145 |
| 3. 1.0 Mg MBM+K+N$_{79}$ ** | 158 $_{(79 + 79)}$ | 45 | 145 | 158 $_{(79 + 79)}$ | 45 | 145 |
| 4. 1.5 Mg MBM+K+N$_{40}$ *** | 158 $_{(118 + 40)}$ | 68 | 145 | 158 $_{(118 + 40)}$ | 68 | 145 |
| 5. 2.0 Mg MBM+K **** | 158 | 90 | 145 | 158 | 90 | 145 |

* Inorganic NPK—mineral fertilization; ** MBM+K+N79—meat and bone meal with mineral potassium (145 kg K ha$^{-1}$) and nitrogen (79 kg N ha$^{-1}$) fertilizers; *** MBM+K+N40—meat and bone meal with mineral potassium (145 kg K ha$^{-1}$) and nitrogen (40 kg N ha$^{-1}$) fertilizers; **** MBM+K—meat and bone meal with mineral potassium (145 kg K ha$^{-1}$) fertilizer.

In the NPK treatment (2), only mineral fertilizers were applied: nitrogen (N)—158, phosphorus (P)—45 and potassium (K)—145 kg ha$^{-1}$. Part of the N fertilizer (30 kg ha$^{-1}$) was applied presowing in the form of urea (46% N), and the remainder was top-dressed twice: 80 kg N ha$^{-1}$ at the beginning of the growing season and 48 kg N ha$^{-1}$ at the beginning of the bud formation stage, in the form of ammonium nitrate (34% N). Phosphorus was applied presowing at 45 kg ha$^{-1}$ in the form of granular triple superphosphate (20.1% P). Potassium was applied presowing at a total rate of 145 kg K ha$^{-1}$, as two fertilizers—potassium chloride (49.8% K) and potassium sulfate (41.5% K and 17% S), at 72 and 73 kg K ha$^{-1}$, respectively. Sulfur (S) was applied with potassium sulfate, at 30 kg ha$^{-1}$.

In order to widen a too-narrow N:P ratio in MBM (1 Mg contains 79 kg N and 45 kg P), supplemental mineral N was applied in two treatments (3 and 4) as ammonium nitrate (34% N), at 79 and 40 kg, as part of the total N rate (158 kg ha$^{-1}$) applied in all treatments (except for treatment 1—without

fertilization) (Table 1). Ammonium nitrate was applied in spring, at the beginning of the growing season. Mineral N accounted for 50% and 25% of the total N rate in treatments 3 and 4, respectively. Phosphorus was applied presowing as MBM (1.0, 1.5 and 2.0 Mg ha$^{-1}$), at 45, 68 and 90 kg ha$^{-1}$. MBM had a low K content (3.3 kg K per 1.0 Mg)—the K rate was 145 kg ha$^{-1}$ in all treatments—therefore it was applied presowing with mineral fertilizers (potassium chloride and potassium sulfate), as in the NPK treatment. The MBM used in the experiment contained (per 1 kg dry matter (DM)): 96.3% DM, 710 g organic matter, 280 g crude ash, 137 g crude fat, 78.7 g N, 45.3 g P, 3.32 g K, 100.1 g Ca, 6.8 g Na and 2.0 g Mg kg$^{-1}$. Its pH in H$_2$O was 6.3. The MBM was a low-risk (category 3) material, purchased from the Animal By-Products Disposal Plant SARIA Poland in Długi Borek near Szczytno.

## 2.2. Experimental Design and Crop Management

The experiment was established on brown soil developed from loamy sand, Dystric Cambisol according to the World Reference BaseWRB [25], and it had a randomized block design with four replications. The soil was slightly acidic (pH in 1 M KCl = 5.61), it had a mineral N content of 8.82 mg kg$^{-1}$, and it was moderately abundant in available P (65 mg kg$^{-1}$), abundant in K (163 mg K kg$^{-1}$) and highly abundant in Mg (96 mg kg$^{-1}$ soil).

Winter oilseed rape of the hybrid cultivar 'SY SAVEO' was sown each year on 25–26 August, at a density of 50 germinating seeds per m$^2$ and a row spacing of 20 cm. Harvested plot size was 20 m$^2$ (4 × 5 m). The preceding crop was winter wheat (*Triticum aestivum* L.). All cultivation and crop protection measures were applied in accordance with good agricultural practices. In each plot, the seed yield of winter oilseed rape was determined after threshing in terms of weight, and adjusted to standard moisture content (9%). The results were expressed in terms of 1 ha. Thousand seed weight (TSW) (3 × 300 seeds per plot) was calculated at standard moisture content (9%).

## 2.3. Chemical Composition of Plants

Plant samples were dried, weighted, ground and wet mineralized in concentrated sulfuric (VI) acid with hydrogen peroxide (H$_2$O$_2$) as the oxidizing agent. Seeds of winter oilseed rape were assayed for the content of total N, crude fat, GLS and fatty acid profile. Total N content was determined by the hypochlorite method, and protein concentration was estimated by multiplying total N by conversion factor 6.25. Crude fat content was determined by Soxhlet extraction with the use of petroleum ether (analytical grade). Fatty acids were separated using a VARIAN CP-3800 gas chromatograph. The concentrations of acid detergent fiber (ADF), neutral detergent fiber (NDF) and GLS were determined by near infrared spectroscopy (FOSS NIR Systems Inc., Silver Spring, MD, USA) [8,26].

## 2.4. Statistical Analysis

The results were processed statistically by repeated measures analysis of variance (ANOVA) using STATISTICA 12 software [27], where MBM dose was the fixed grouping factor (5 fertilization treatments), and year of the study was the repeated measurement factor (two years). The significance of differences between mean values was estimated by Tukey's test at a significance level of $p < 0.05$.

## 2.5. Weather Conditions

The experiment was performed under adverse weather conditions (Figure 1). In the first year of the study, winter oilseed rape was sown at the optimum date (26 August 2015). In August and September, mean air temperature was 1.9 °C and 0.7 °C higher than the long-term average (1981–2010). Precipitation was uneven and differed considerably from the long-term average. Drought persisted in August, which adversely affected seeding and seed germination. Precipitation was also two-fold lower than the long-term average in October. November and December were warmer and wetter. The period of winter dormancy in 2015/2016 considerably differed from the long-term pattern. January was frosty, with no frost cover. In February and March, mean air temperatures were substantially

higher (mostly above zero), and precipitation in February was over two-fold higher than the long-term average. Mean air temperature in March was 1.2 °C higher than the long-term average, but ground frost occurred between 10 and 20 March. In subsequent months (until harvest), mean air temperatures were optimal, but abundant rainfall in July (which exceeded the long-term average 1.8-fold) hindered harvest operations, which were performed on 18 July 2016.

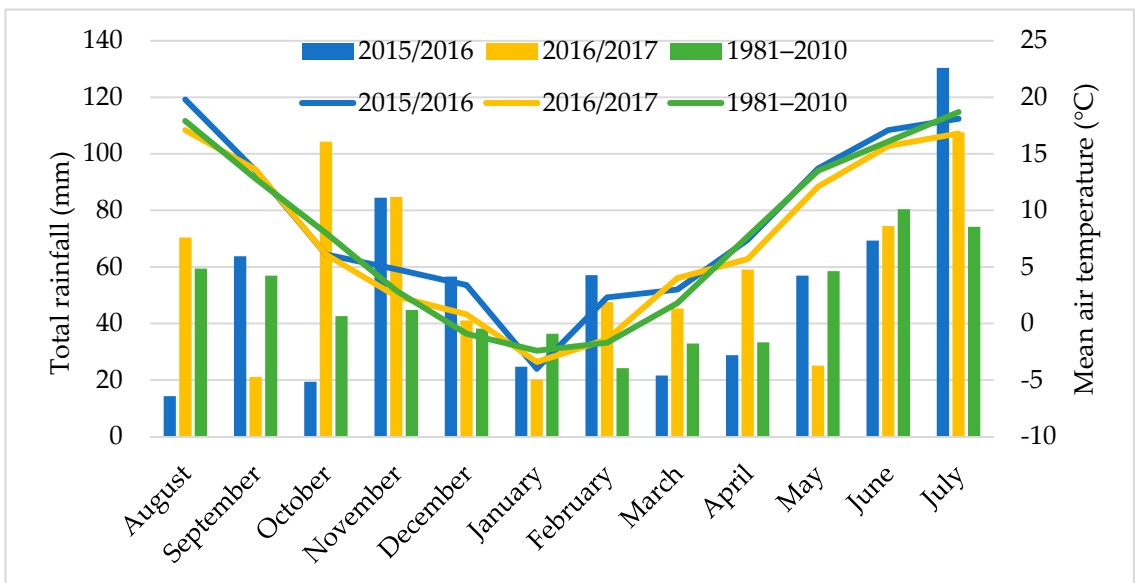

**Figure 1.** Weather conditions in 2015–2017, and in the 1981–2010 reference period, according to the Agricultural Experiment Station in Tomaszkowo (Bars refer to total rainfall and lines refer to mean air temperature).

In the second growing season (2016/2017), mean air temperatures were similar to the long-term average, whereas total precipitation levels were considerably higher (by 119.3 mm) than the long-term average of 1981–2010, and unevenly distributed (Figure 1). Winter oilseed rape was sown on 25 August 2016. In autumn, a dry spell was noted only in September (precipitation was 2.7-fold lower than the long-term average), whereas in October and November, rainfall substantially exceeded the water needs of plants. In spring, rainfall deficiency was observed in May (precipitation was 2.3-fold lower than the long-term average), which reduced rapeseed yields. Similarly to the first year of the study, abundant precipitation in July (1.5-fold higher than the long-term average), hindered harvest which was carried out on 20 July 2017.

## 3. Results and Discussion

### 3.1. Seed Yield

The field experiment revealed that the seed yield of winter oilseed rape was affected by fertilization and weather conditions (Table 2). Over the two-year study, the average seed yield ranged from 1.71 Mg ha$^{-1}$ (zero-fert.) to 3.14 Mg ha$^{-1}$ (inorganic NPK treatment). The yields of winter oilseed rape fertilized with MBM were significantly lower (by approx. 0.5 Mg ha$^{-1}$ seeds on average) than in the NPK treatment, but higher than in the unfertilized treatment (by approx. 0.9 Mg ha$^{-1}$ seeds on average). In MBM treatments, seed yields were comparable regardless of the rate of supplemental mineral N, ranging from 2.49 to 2.69 Mg ha$^{-1}$. The observed increase in oilseed rape yields can be attributed not only to breeding progress and modern production technology, but also to higher agricultural inputs. Fertilization continues to be the main yield-forming factor in the cultivation of winter oilseed rape [1,3,5,6,8,26,28,29]. The use of MBM as a substitute for mineral fertilizers provides both economic and environmental benefits [12,16–18].

**Table 2.** Seed yield and quality of winter oilseed rape.

| Treatments | Seed Yield (Mg ha$^{-1}$, 91% DM) | TSW * (g, 91% DM) | Protein Content (g kg$^{-1}$ DM) | Protein Yield (Mg ha$^{-1}$) | Fat Content (g kg$^{-1}$ DM) | Fat Yield (Mg ha$^{-1}$) | ADF ** (%) | NDF *** (%) |
|---|---|---|---|---|---|---|---|---|
| 1. Zero-fert | 1.71 $^a$ | 4.96 $^a$ | 177 $^a$ | 0.27 $^a$ | 499 $^{bc}$ | 0.77 $^a$ | 23.3 | 29.5 $^{ab}$ |
| 2. Inorganic NPK * | 3.14 $^c$ | 5.23 $^{ab}$ | 198 $^b$ | 0.56 $^c$ | 479 $^a$ | 1.35 $^c$ | 22.1 | 27.7 $^a$ |
| 3. 1.0 Mg MBM+K+N$_{79}$ | 2.65 $^b$ | 5.21 $^{ab}$ | 189 $^{ab}$ | 0.45 $^b$ | 486 $^{ab}$ | 1.16 $^b$ | 22.7 | 28.5 $^{ab}$ |
| 4. 1.5 Mg MBM+K+N$_{40}$ | 2.69 $^b$ | 5.32 $^b$ | 178 $^a$ | 0.43 $^b$ | 502 $^c$ | 1.22 $^{bc}$ | 23.5 | 29.7 $^b$ |
| 5. 2.0 Mg MBM+K | 2.49 $^b$ | 5.11 $^{ab}$ | 186 $^{ab}$ | 0.41 $^b$ | 493 $^{bc}$ | 1.11 $^b$ | 23.1 | 28.8 $^{ab}$ |
| Annual mean 2016 | 2.67 $^B$ | 5.24 $^B$ | 189 $^B$ | 0.46 $^B$ | 489 $^A$ | 1.17 $^B$ | 21.8 $^A$ | 27.8 $^A$ |
| Annual mean 2017 | 2.41 $^A$ | 5.10 $^A$ | 182 $^A$ | 0.40 $^A$ | 495 $^B$ | 1.07 $^A$ | 24.1 $^B$ | 29.9 $^B$ |
| Interaction (t × y) | s | ns | ns | s | ns | s | ns | ns |

* TSW—thousand seed weight; ** ADF—acid detergent fiber; *** NDF—neutral detergent fiber. Explanations as in Table 1. a, b, c, ab, bc—significant differences between means for fertilization (in columns), A, B—significant differences between means for the years 2016 and 2017 (in columns), according to Tukey's test ($p < 0.05$). Interaction between fertilization and year (t × y); s—significant; ns—not significant.

Weather conditions exerted a significant influenced the seed yield of winter oilseed rape (Table 2), which was significantly higher (by 0.26 Mg ha$^{-1}$) in the first year of the study (2016) than in the second year (2017). The differences in seed yield resulted from uneven precipitation, in particular rainfall deficiency in May 2017 when precipitation was 2.3-fold lower than the long-term average (Figure 1). According to Weymann et al. [30], the seed yield of oilseed rape is determined by weather conditions during the growing season by approximately 40%. In our previous study [21], the seed yield of winter oilseed rape fertilized with MBM ranged from 3.7 to 4.2 Mg ha$^{-1}$. Although NPK increased yields by 84% over the unfertilized check, MBM without supplemental N also successfully increased yields by 46%. Stępień and Wojtkowiak [10] also noted high seed yields (4.6–5.3 Mg ha$^{-1}$) in winter oilseed rape fertilized with similar doses of MBM. The seed yield of winter oilseed rape is also determined by factors other than fertilization, such as cultivar, agronomic and environmental conditions [3,9].

The TSW of winter oilseed rape ranged from 4.96 g to 5.32 g on average (Table 2). In comparison with the unfertilized treatment, a significant increase in TSW (by 0.36 g) was observed in the treatment with 1.5 Mg ha$^{-1}$ MBM + 40 kg ha$^{-1}$ mineral N (25% N from mineral fertilizers and 75% N from MBM). It should be noted that in the above treatment, winter oilseed rape was supplied with 68 kg ha$^{-1}$ P from MBM, which appears to be an optimal P rate for seed development. On the other hand, no significant differences in TSW were found between treatments 2, 3, 4 and 5 where increasing P rates were applied (45, 45, 68 and 90 kg P ha$^{-1}$, respectively). Stępień et al. [31] reported similar TSW in winter oilseed rape, in the range of 4.58 to 5.57 g. Thousand seed weight was significantly higher (by 0.14 g on average) in the first year of the study than in the second year.

### 3.2. Total Protein Content and Yield

Over the two-year study, the total protein content of oilseed rape seeds ranged from 177 to 198 g kg$^{-1}$ DM, depending on fertilization (Table 2). Winter oilseed rape had a higher seed protein concentration in the NPK treatment than in the 1.5 Mg MBM ha$^{-1}$ + 40 kg N ha$^{-1}$ treatment and the unfertilized treatment. It should be noted that the total N rate (mineral fertilizers + MBM) was 158 kg ha$^{-1}$ in all treatments. It appears that N from mineral fertilizers was most efficiently utilized by winter oilseed rape, although seeds harvested in MBM treatments (1.0 Mg ha$^{-1}$ MBM + 79 kg N and 2.0 Mg ha$^{-1}$ MBM without supplemental N) had protein content comparable with that noted in the NPK treatment. During the two-year experiment, average protein yield per ha varied widely from 0.27 to 0.56 Mg, depending on fertilization. The seed yield of winter oilseed rape is reflected in protein yield [10,21,31]. Protein yield was significantly higher in the NPK treatment than in the remaining treatments. However, winter oilseed rape fertilized with MBM, regardless of the rate of supplemental mineral N, was characterized by significantly higher protein yield (by 0.16 Mg ha$^{-1}$ on average) relative to the unfertilized treatment. Both the protein content of seeds and biological

yield were significantly modified by weather conditions. In the first year of the study, seed yield was higher, and seeds had a higher protein content, which was reflected in a significantly higher protein yield (by 60 kg ha$^{-1}$), compared with the second year. In a study by Nogala-Kałucka et al. [9], the average protein content of seeds in several cultivars of winter oilseed rape grown in five Polish regions under different environmental and soil conditions was 200 g kg$^{-1}$ DM, which is comparable with the present results.

### 3.3. Crude Fat Content and Yield

The crude fat content of oilseed rape seeds ranged from 479 to 502 g kg$^{-1}$ DM (Table 2). Seeds harvested in the 1.5 Mg MBM ha$^{-1}$ + 40 kg ha$^{-1}$ N treatment had a significantly higher crude fat content than seeds harvested in the NPK treatment and the 1.0 Mg MBM ha$^{-1}$ + 79 kg ha$^{-1}$ N treatment. It appears that the higher rate of supplemental mineral N (79 kg ha$^{-1}$) contributed to a decrease in the crude fat content of seeds and an increase in the total protein content of seeds. Higher levels of N fertilization usually lead to a decrease in crude fat concentration and a simultaneous increase in total protein concentration in the seeds of winter oilseed rape [5,29,32,33]. Seeds of the currently cultivated winter oilseed rape varieties contain approximately 400–420 g kg$^{-1}$ DM fat and 220–235 g kg$^{-1}$ DM protein [34]. Winter oilseed rape of the hybrid cultivar 'SY SAVEO' contains 480 g kg$^{-1}$ DM fat on average [35]. In the present study, the seeds of winter oilseed rape had a lower protein content and higher crude fat content.

Fat yield varied widely from 0.77 to 1.35 Mg ha$^{-1}$, depending on fertilization (Table 2). The highest fat yield per ha was achieved in the NPK treatment (1.35 Mg ha$^{-1}$) and in the 1.5 Mg MBM ha$^{-1}$ + 40 kg N treatment (1.22 Mg ha$^{-1}$). The highest yield of seeds with the lowest crude fat content was noted in the NPK treatment, whereas a significantly lower yield of seeds with higher crude fat content was observed in MBM treatments. Both the crude fat content of seeds and fat yield were determined by weather conditions. The crude fat content of seeds was significantly lower in the first year of the study than in the second year. However, fat yield was significantly higher (by 0.10 Mg ha$^{-1}$) in the first year, due to the higher seed yield. According to Chmura et al. [32], high precipitation levels and high air temperatures in December–March and April–May, and low precipitation levels (25 mm) and high air temperatures (around 19 °C) from the end of flowering to processing maturity contribute to a high crude fat content of seeds in winter oilseed rape. Such a relationship was not observed in the current study. In the first year of the study, mean air temperatures were high between December–March and flowering, but precipitation was lower than in the second year. In both years of the study, abundant rainfall in July hindered harvest operations (Figure 1).

### 3.4. Fiber Fractions

Mineral fertilization usually leads to an increase in the concentrations of NDF and ADF in the seeds of winter oilseed rape [26,29] and other Brassica crops [7,36]. In the present study, only the NDF content of seeds was significantly affected by fertilization (Table 2). Seeds harvested in the NPK treatment had the lowest NDF content (27.7%). A significant increase in the NDF content of seeds, to 29.5% and 28.5–29.7%, was noted in the unfertilized treatment and MBM treatments (regardless of dose), respectively. The concentrations of ADF and NDF were higher (by 2.3% and 2.1%, respectively) in the second year of the study, characterized by relatively low mean daily air temperatures during seed ripening (0.4 to 1.9 °C lower than the long-term average). An increase in the concentrations of both fiber fractions compromises the feed value of non-fat seed residues: an increase in ADF content decreases nutrient digestibility, and an increase in ADF content decreases feed utilization by animals [37].

### 3.5. Fatty Acid Profile

Rapeseed oil has high biological and nutritional value due to the presence of fatty acids with various degrees of unsaturation and desirable ratios. It also contains small amounts of saturated fatty acids [38]. The concentrations of 14 fatty acids, including seven saturated fatty acids, five monounsaturated

fatty acids and two polyunsaturated fatty acids, were determined in the seeds of winter oilseed rape (Table 3).

**Table 3.** Fatty acid profile of rapeseed oil (%).

| Treatments | | C16:0 * | C18:0 * | C18:1 * | C18:2 * | C18:3 * | C20:0 * | C20:1 * | C22:1 * | Other ** | C18:2/C18:3 |
|---|---|---|---|---|---|---|---|---|---|---|---|
| 1. Zero-fert | | 5.88 | 1.99 | 56.65 | 20.97 | 11.67 | 0.68 [a] | 1.18 [a] | 0.03 | 0.95 | 1.81 |
| 2. Inorganic NPK * | | 5.87 | 2.13 | 56.67 | 20.94 | 11.54 | 0.71 [b] | 1.26 [c] | 0.02 | 0.99 | 1.83 |
| 3. 1.0 Mg MBM+K+$N_{79}$ | | 5.87 | 1.96 | 57.06 | 20.90 | 11.30 | 0.69 [ab] | 1.23 [bc] | 0.03 | 0.97 | 1.86 |
| 4. 1.5 Mg MBM+K+$N_{40}$ | | 5.86 | 1.87 | 56.42 | 21.02 | 11.96 | 0.68 [a] | 1.20 [ab] | 0.03 | 0.96 | 1.76 |
| 5. 2.0 Mg MBM+K | | 5.85 | 2.11 | 56.65 | 21.05 | 11.68 | 0.69 [ab] | 1.22 [b] | 0.02 | 0.98 | 1.81 |
| Annual mean | 2016 | 5.90 [B] | 2.10 | 55.91 [A] | 22.09 [B] | 11.15 [A] | 0.71 [B] | 1.24 [B] | 0.02 | 0.98 | 1.98 [B] |
| | 2017 | 5.83 [A] | 1.93 | 57.47 [B] | 19.86 [A] | 12.11 [B] | 0.67 [A] | 1.19 [A] | 0.03 | 0.96 | 1.64 [A] |

\* C16:0—palmitic acid, C18:0—stearic acid, C18:1—oleic acid, C18:2—linoleic acid, C18:3—α-linolenic acid, C20:0—arachidic acid, C20:1—eicosenoic acid, C22:1—erucic acid; ** total content of other fatty acids: C14:0 (myristic) + C15:0 (pentadecanoic) + C16:1 (palmitoleic) + C17:0 (margaric) + C17:1 (margarine oleic) + C22:0 (behenic). a, b, c, ab, bc—significant differences between means for fertilization (in columns), A, B—significant differences between means for the years 2016 and 2017 (in columns), according to Tukey's test ($p < 0.05$).

A statistical analysis of the fatty acid composition of rapeseed oil revealed minor changes in the concentrations of monoenoic and polyenoic fatty acids. Fertilization had a significant effect on the content of arachidic acid C20:0 and eicosenoic (gadoleic) acid C20:1, which was highest in seeds harvested in the NPK treatment. The major unsaturated fatty acids with 18 carbon atoms (oleic acid C18:1, linoleic acid C18:2 and α-linolenic acid C18:3) accounted for nearly 90% of total fatty acids, and their proportions were not modified by fertilization. Oil extracted from oilseed rape seeds contains mainly oleic acid (60–65%), linoleic acid (18.5–20.0%) and α-linolenic acid (7–11%), and its quality is superior to that of oils obtained from other oil-bearing seeds [9,38]. According to Molazem et al. [39], the average content of saturated and unsaturated fatty acids in oilseed rape seeds is 7% and 61%, respectively. Such a fatty acid profile is recognized by nutritionists as perfect for human consumption [38]. From the nutritional perspective, an important role is played not only by the concentrations of fatty acids, but also by their ratios. This applies in particular to essential unsaturated fatty acids that cannot be synthesized in the human body, including linoleic acid C18:2 and α-linolenic acid C18:3. Averaged across treatments and years, the rapeseed oil in our study was found a have a linoleic acid to α-linolenic acid ratio of 1.81:1 which is considered to be a stable ratio. Significant differences in the values of this ratio (from 1.64 to 1.98) were observed between the years of the study. The dietary linoleic acid to α-linolenic acid ratio of 2:1 seems to be ideal in view of the important role played by both acids in the human body [9]. Tańska et al. [38] and Stępień et al. [31] reported a wider ratio of linoleic acid to α-linolenic acid. In the current study, rapeseed oil contained trace amounts (0.025% on average) of erucic acid C22:1, and the total content of other fatty acids, i.e., C14:0 (myristic), C15:0 (pentadecanoic), C16:1 (palmitoleic), C17:0 (margaric), C17:1 (margarine oleic) and C22:0 (behenic), was below 1.0%. Similar values were reported by Stępień et al. [31]. The proportions of both unsaturated and saturated fatty acids in rapeseed oil were modified by weather conditions to a greater extent than by fertilization. The seeds of winter oilseed rape harvested in the first year of the study had significantly higher concentrations of palmitic acid (C16:0), linoleic acid (C18:2), arachidic acid (C20:0) and gadoleic acid (C20:1) than those harvested in the second year. The opposite trend was noted for the content of oleic acid (C18:1) and α-linolenic acid (C18:3). Nogala-Kałucka et al. [9] found no significant differences in the concentrations of oleic acid, linoleic acid or α-linolenic acid in rapeseed oil extracted from seeds harvested in five Polish regions with different environmental and soil conditions; significant differences were observed only in the content of gadoleic acid.

### 3.6. Glucosinolate Content

The canola varieties of winter oilseed rape included in the Polish National List of Agricultural Plant Varieties contain approximately 9.1 to 14.7 μmol GLS $g^{-1}$ seeds [39]. The seeds of double-low varieties of

winter oilseed rape contain mostly progoitrin, gluconapin (alkenyl GLS) and 4-hydroxyglucobrassicin (indole GLS), which account for approximately 90–91% of total GLS [40]. The types and concentrations of GLS in crops are influenced by genetic factors as well as environmental and soil conditions [41], and agronomic practices [5,8,26,29,40]. In the group of agronomic factors, fertilization plays a key role in GLS biosynthesis [5]. In the present study, the seeds of winter oilseed rape in the unfertilized treatment had the lowest content of total GLS and progoitrin, which is highly desirable (Table 4).

**Table 4.** Glucosinolate content of winter oilseed rape seeds ($\mu$mol g$^{-1}$ DM).

| Treatments | | Glucosinolates | | | |
|---|---|---|---|---|---|
| | | Total | Gluconapin | Progoitrin | 4-Hydroxyglucobrassicin |
| 1. Zero-fert | | 10.33 [a] | 2.18 | 4.02 [a] | 3.23 |
| 2. Inorganic NPK | | 11.50 [ab] | 2.35 | 4.91 [b] | 3.40 |
| 3. 1.0 Mg MBM+K+N$_{79}$ | | 11.79 [b] | 2.29 | 5.10 [b] | 3.50 |
| 4. 1.5 Mg MBM+K+N$_{40}$ | | 11.21 [ab] | 2.25 | 4.65 [ab] | 3.43 |
| 5. 2.0 Mg MBM+K | | 11.61 [ab] | 2.26 | 5.01 [b] | 3.42 |
| Annual mean | 2016 | 10.82 [A] | 1.98 [A] | 4.28 [A] | 3.72 [B] |
| | 2017 | 11.76 [B] | 2.55 [B] | 5.20 [B] | 3.07 [A] |
| Interaction (t × y) | | s | ns | s | s |

a, b, ab—significant differences between means for fertilization (in columns), A, B—significant differences between means for the years 2016 and 2017 (in columns), according to Tukey's test ($p < 0.05$). Interaction between fertilization and year (t × y); s—significant, ns—not significant.

The application of mineral fertilizers and MBM caused an increase in the content of total GLS and progoitrin by 0.88–1.46 and 0.63–1.08 $\mu$mol g$^{-1}$ DM, respectively. The seeds of winter oilseed rape in the 1.0 Mg MBM ha$^{-1}$ + 79 kg ha$^{-1}$ N treatment had the significantly highest content of total GLS and progoitrin. Jankowski et al. [1] and Jankowski et al. [8] also demonstrated that higher rates of mineral fertilizers increased the concentrations of gluconapin, progoitrin (alkenyl GLS) and 4-hydroxyglucobrassicin (indole GLS) by 2.5-, 1.7- and 1.1-fold, respectively. Yang et al. [42], Jankowski et al. [29] and Groth et al. [5] found that the feed value of non-fat seed residues improved (the concentrations of total GLS, including alkenyl GLS, decreased) in response to increased fertilization levels. In the current study, the seeds of winter oilseed rape harvested in the first year of the study were characterized by a considerably higher feed value (low content of gluconapin and progoitrin, high content of 4-hydroxyglucobrassicin). In the second year, abundant rainfall and relatively low mean daily temperatures during seed ripening promoted the biosynthesis of alkenyl GLS (gluconapin and progoitrin), which limited the feed value of seeds, and suppressed the biosynthesis of the desirable indole GLS (4-hydroxyglucobrassicin).

## 4. Conclusions

The yields of winter oilseed rape fertilized with MBM were significantly lower than in the NPK treatment, but higher than in the unfertilized treatment. Neither seed yield nor quality were affected by the rates of supplemental mineral N (79 and 40 kg ha$^{-1}$) in treatments with MBM and the total N rate of 158 kg ha$^{-1}$. The average protein and crude fat content of seeds was 186 g kg$^{-1}$ DM and 492 g kg$^{-1}$ DM, respectively. Winter oilseed rape accumulated significantly more protein in seeds in the NPK treatment than in the 1.5 Mg ha$^{-1}$ MBM + 40 kg N ha$^{-1}$ treatment and the unfertilized treatment. The crude fat content of seeds was significantly higher in the 1.5 Mg MBM ha$^{-1}$ + 40 kg N ha$^{-1}$ treatment, compared with the NPK treatment and the 1.0 Mg MBM ha$^{-1}$ + 79 kg N ha$^{-1}$ treatment. Fertilization had a minor influence on the proportions of fatty acids in rapeseed oil, except for arachidic acid and gadoleic acid. The major unsaturated fatty acids with 18 carbon atoms (oleic, linoleic, and $\alpha$-linolenic acids) accounted for nearly 90% of total fatty acids, and the average ratio of linoleic acid to $\alpha$-linolenic acid was found to be 1.81:1. Differences between years in all of the traits appeared to be related to greater stress during the seed development period of year two. The seeds of winter oilseed rape harvested in

the first year were characterized by higher feed value, i.e., low concentrations of gluconapin, progoitrin, ADF and NDF, and a high content of 4-hydroxyglucobrassicin. Seeds harvested in the second year had a higher crude fat content.

**Author Contributions:** Conceptualization, A.N. and A.Z.; methodology, A.N. and A.Z.; chemical analyses, A.Z.; data analysis, A.N. and A.Z.; investigation, A.N. and A.Z.; resources, A.N. and A.Z.; data curation, A.N. and A.Z.; writing, A.Z.; writing—review and editing, A.N. and A.Z.; visualization, A.Z.; supervision, A.N; funding acquisition, A.N. All authors have read and agreed to the published version of the manuscript.

**Funding:** This research was supported by the Ministry of Science and Higher Education of Poland as part of statutory activities (No. 20.610.002-110). The project was financially supported by the Minister of Science and Higher Education in the range of the program entitled "Regional Initiative of Excellence" for the years 2019–2022, Project No. 010/RID/2018/19, amount of founding 12.000.000 PLN.

**Conflicts of Interest:** The authors declare no conflict of interest.

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
