# Peer review of "The Effect of Meat and Bone Meal (MBM) on the Seed Yield and Quality of Winter Oilseed Rape"

_agronomy, doi:10.3390/agronomy10121952_

Round 1
Reviewer 1 Report
Review of Zaluszniewska and Nogalska, Effects of Meat-Bone Meal on Oilseed Rape
Overall, this manuscript is very good. It is organized well and the authors provided some important findings on how MBM and NPK affect oilseed rape yield and seed quality. My only comments are editorial/clerical and relate to making the manuscript clearer in certain sections.
Abstract
- The abstract needs to state that this study was conducted for two seasons.
- The abstract needs to state the methods information in its own separate sentence rather than trying to combine them with the results. Try “Five treatments were compared, MBM applied at 1.0, 1.5, 2.0 Mg/ha, inorganic NPK, and a zero-N check. All three MBM plots and the NPK plots received supplemental inorganic N to provide a total of 158 kg N/ha.”
- “Neither MBM doses (1.0, 1.5 and 2.0 Mg/ha) nor the rates of supplemental mineral N (40 and 79 kg N/ha) affected the yields of seeds, protein or fat” is poorly worded. This sentences needs rewritten. Try “All plots receiving MBM yielded equally to each other but greater than the unfertilized check.”
- I don’t think it is a good idea to call the NPK treatment the “control.” It gets confused with the unfertilized control check treatment. I understand what the author are trying to convey here by using “control” but the NPK treatment can be called many other names. How about the “full inorganic NPK treatment” or the “conventional fertilizer treatment”?
Introduction
- The following is a confusing sentence. “Nitrogen supplied by MBM can meet the N requirements of cereals in 80% [13] and the remainder (20%) must be provided by mineral fertilizers.” Try “Nitrogen supplied by MBM can meet 80% of the N requirements of cereals [13] but the remaining 20% must be provided by mineral fertilizers.”
- This sentence is also confusing “… that lower doses of MBM applied with supplemental mineral N to widen the narrow N:P ratio in MBM…” Try “ ….. that lower doses of MBM supplemented with mineral N might raise the soil N:P ratio from 1.75 to values above 2.0.”
Materials and Methods
- Instead of the letter ‘O’ for the zero fert treatment, why not label it “Zero-Fert”? It is also listed as Treatment #1 so also calling it ‘O’ unnecessarily adds more to the confusion.
- When describing the concentration of elements in MBM, do not use per kg‑1, just use per or kg‑1 not both. Otherwise it is redundant.
- I did not see pre-season soil N levels presented although the authors did provided values for pH, P, K, and Mg.
- Figure 1 does not specifically indicate that the bars refer to rainfall and that the lines refer to temperature. I think for those of us that have been around for a while understand this but the figure caption needs to add “Bars refer to rainfall and lines refer to temperature.”
- Figure 1 has a blue legend that reads “20116/17.” Need to remove a “1.”
- The figure legends are confusing. Why is there a blue line for temperature in 2015-2016 and then why is blue used again for rainfall for the second year (2016-2017)? Same problem with the orange color. The appear flip-flopped. Very confusing. Why not match blue to year one and orange to year two?
Results and Discussion
- The authors mention that the difference in yield between years was significant and even put letters next to the year-average values. The statistical test comparing years is usually not very valid (no real replicates on years). So maybe removing the word significant is the way to handle that. Or putting a caveat in the Table footnote about this.
- Following up on #1, the authors do not mention whether there was a year-by-treatment interaction on yield but there probably was (my experience has been that it happens for every experiment). Regardless, the P-value for the year-by-treatment interaction could also be placed in a footnote within Table 2. Maybe adding two more rows in Table 2, add the Year effects P-value and the Year-by-treatment P-value.
- Write “Although NPK increased yields by 84% over the unfertilized check, MBM without supplemental N also successfully increased yields by 46%” instead of what was written “It should be stressed that MBM applied at 1.0 to 2.5 Mg/ha without supplemental mineral N was sufficient to considerably increase seed yields.”
- The authors write “On the other hand, no significant differences in seed quality were found between treatments 2, 3, 4 and 5 where increasing P rates were applied (45, 45, 68 and 90 kg P/ha, respectively).“ Do the authors mean TKW instead of “seed quality”? I see differences in protein and fat concentration so “seed quality” are not the correct words.
- When distinguishing protein concentration and protein yield, the authors need to be more careful with their words. Revise the following sentence: Winter oilseed rape accumulated significantly more protein in seeds in the NPK treatment than in the 1.5 Mg ha-1 MBM + 40 kg N ha-1 treatment and the unfertilized treatment. Try: Winter oilseed rape had a higher seed protein concentration in the NPK treatment than in the 1.5 Mg MBM/ha + 40 kg N/ha treatment (Trt 4) and the unfertilized treatment (Trt 1).
- The values of protein and fat concentration in Table 2 need to be rounded off to only three digits. I doubt the precision of those values is really wxy.z as opposed to simply wxy without a value after the decimal.
- Are the authors concerned that the fat concentrations of their seeds are 20% higher than the expect values from the literature? The authors report very high fat concentrations and readers are going to be interested in and skeptical of that. Are there data available for fat concentrations of SY SAVEO?
- I do not understand what this sentence means “In both years of the study, the total amount of rainfall between flowering and harvest exceeded the optimal level (25 mm) several times (Figure 1).” Please revise.
- The first paragraph of the section for Fatty Acid Profile has some author notes or guide to authors or something like that. It needs to be deleted.
- Change this sentence: The analyzed rapeseed oil was characterized by a stable ratio of linoleic acid to α-linolenic acid, which was determined at 1.81:1 on average” to “Averaged across treatments and years, the rapeseed oil in our study was found a have a linoleic acid to α‑linolenic acid ratio of 1.81:1 which is considered to be a stable ratio.“
- I do not understand the units on GLS μM/g sm? Do the authors mean micromol(e) per g? That would be μmol/g not capital M. Capital M indicates molarity to many of us.
Conclusion
- Sentence needs revised. The major unsaturated fatty acids with 18 carbon atoms (oleic, linoleic and α-linolenic acids) accounted for nearly 90% of total fatty acids, and the average ratio of linoleic acid to α-linolenic acid remained stable at 1.81:1. I would replace “remained stable” with “was found to be.”
- Delete or reword the conclusions about years. “The yields of high-quality seeds, protein and fat were significantly higher in the first year than in the second year of the study.” As mentioned earlier, years are almost always different and there is no way to repeat the weather that occurred in 2015-2107. Perhaps write: “Differences between years in all of the traits appeared to be related to greater stress during the seed development period of year two.”
Author Response
Dear Reviewer Thank you for positively assessing the article. I am sending replies to all comments in the attached file. Yours faithfullyAleksandra Załuszniewska

Reviewer 2 Report
The manuscript "The effect of meat and bone meal (MBM) on the seed yield and quality of winter oilseed rape" evaluate the effect of meat and bone meal applied without or with mineral nitrogen on yield parameters and chemical composition of winter oilseed rape (Brassica napus L.). The proposed article is of interest in the field and studied many aspects related to chemical composition of rapeseed oil. It is a well conducted study.
Author Response
Dear Reviewer
Thank you very much for the time and effort devoted to reviewing the manuscript. Thank you very much for the positive feedback. Your help and cooperation are highly appreciated.
Best Regards
Aleksandra Załuszniewska